# Transgene Was Silenced in Hybrids between Transgenic Herbicide-Resistant Crops and Their Wild Relatives Utilizing Alien Chromosomes

**DOI:** 10.3390/plants11233187

**Published:** 2022-11-22

**Authors:** Zicheng Shao, Lei Huang, Yuchi Zhang, Sheng Qiang, Xiaoling Song

**Affiliations:** Weed Research Laboratory, College of Life Sciences, Nanjing Agricultural University, Nanjing 210095, China; shaozicheng1994512@163.com (Z.S.); 2020116015@stu.njau.edu.cn (L.H.); 2014116095@njau.edu.cn (Y.Z.); wrl@njau.edu.cn (S.Q.)

**Keywords:** *Brassica juncea*, *Brassica napus*, gene silencing, methylation, alien chromosomes, transgenes

## Abstract

The commercialization of transgenic herbicide-resistant (HR) crops may cause gene flow risk. If a transgene in progenies of transgenic crops and wild relatives is silencing, these progenies should be killed by the target herbicide, thus, the gene flow risk could be decreased. We obtained the progenies of backcross generations between wild *Brassca juncea* (AABB, 2n = 36) and glufosinate-resistant transgenic *Brassica napus* (AACC, 2n = 38, *PAT* gene located on the C-chromosome). They carried the HR gene but did not express it normally, i.e., gene silencing occurred. Meanwhile, six to nine methylation sites were found on the promoter of *PAT* in transgene-silencing progenies, while no methylation sites occurred on that in transgene-expressing progenies. In addition, transgene expressing and silencing backcross progenies showed similar fitness with wild *Brassica juncea*. In conclusion, we elaborate on the occurrence of transgene-silencing event in backcross progenies between transgenic crop utilizing alien chromosomes and their wild relatives, and the DNA methylation of the transgene promoter was an important factor leading to gene silencing. The insertion site of the transgene could be considered a strategy to reduce the ecological risk of transgenic crops, and applied to cultivate lower gene flow HR crops in the future.

## 1. Introductions

The possibility of transgene flow from herbicide-resistant crops to its wild relatives, especially weeds, has been widely concerned by agricultural researchers. As one of the most important transgenic crops in the world, transgenic oilseed rape (*Brassica napus*, AACC, 2n = 38), has been convinced to exist multiple wild relatives, also presents characteristics such as long flower season, long pollen flow distance and long-term pollen viability, which benefit its outcross with wild relatives [1]. Successful gene flow, i.e., the introgression, from transgenic crop to wild relatives depends on the initial crop–weed hybridization, fitness of the first and successive generations of hybrids and the likelihood of transgenes or other alleles spreading from crops to related wild populations [2,3,4].

Sexual compatibility between transgenic crop and wild relatives is the main factor affecting gene flow, which is related to the homology of genome, that is, the higher the homology of the chromosome, the higher the seed-setting rate of the hybrids [5,6]. As the typical relative of *Brassica napus*, wild *Brassica juncea* (AABB, 2n = 36), which was wildly distributed in the wastelands and farmlands of the Yangtze River Basin, Yellow River Basin and northwest China, represents a high gene homology with transgenic *Brassica napus* [7]. As a matter of fact, *Brassica napus* really hybridized with wild *Brassica juncea* to produce F1 under natural conditions [8,9,10,11], and the fitness of the progenies displayed increased fecundity and fitness progressively across the self-pollination of backcross generations, which indicated a potential ecological risk of transgene flow [12,13,14]. However, potential ecological risk depends on not only fitness of the hybrids but also genetic stability of the transgene, including its transmission and expression.

Stable inheritance and expression of HR transgenes in hybrids are considered necessary conditions for transgenic introgression [15,16,17,18,19]. If the transgene cannot be naturally inherited or even if it can be inherited but cannot be expressed, these hybrids couldn’t survive under the pressure of target herbicide selection. Subsequently, the transgene couldn’t fulfill its biological functions under the pressure of target herbicide.

Theoretical predictions suggest that introgression from *Brassica napus* to its relatives, *Brassica rapa* or *Brassica juncea*, should occur more easily when the transgene is originally carried by the A-chromosomes rather than the C-chromosomes, as the C-chromosomes have no homologous partners during meiosis [20,21]. The previous research reported that the persistence of the transgene trait from *Brassica napus* to its relatives deviated significantly from Mendelian segregation under herbicide selection, and the most plausible explanation for this was that the transgene located in the A-chromosome had higher genetic stability than that located in the C-chromosome, which was lacking in *Brassica rapa* and *Brassica juncea* [22,23]. However, there is no empirical evidence to describe this internal mechanism. Moreover, it was found that the survival rate of backcross progenies of HR transgenic *Brassica napus* (*PAT* gene insertion in the C-genome) and wild *Brassica juncea* under herbicide selection was significantly lower than that of backcross progenies of resistant transgenic *Brassica napus* (EPSPS gene insertion in the A-genome) and wild *Brassica juncea* [14]. *PAT* is the type of N-acetyltransferase gene which has the detoxification effect of glufosinate. Bayer Crop Science cloned the genes *Bar* and *PAT* that can encode N-acetyltransferase from *Streptomyces hygroscopicus* and *Streptomyces viridochromo*, respectively [11]. This research provided evidence which was in favor of the C-genome as a safer candidate for transformation [14]. However, we do not know whether the transgene cannot be normally transmitted to the progenies or that the transgene cannot be normally expressed after transmission, i.e., gene silencing.

Gene silencing refers to the molecular biological mechanism induced by double-stranded RNA in eukaryotes to inhibit or terminate gene expression and function by the recognition and elimination of abnormal cellular RNA. Two types of gene silencing have been studied by researchers: transcriptional gene silencing (TGS) and post-transcriptional gene silencing (PTGS), both regulated by small interference RNA (siRNA) pathway [24]. TGS mainly occurs in the endogenous transposon and DNA repeat region, while plants mostly trigger s-PTGS (sense transgene induced PTGS) to induce DNA methylation on the promoter by RNA-mediated DNA methylation (RdDM) pathway [25,26]. The latest research showed that TGS mainly depended on the 24 nt siRNA produced by CLSY1 (Domain Containing Protein Class 1)—NRPD1 (DNA Directed PoL IV Subunit1) in the RdDM pathway, while PTGS depends on the methylation process mediated by NRPE1 (DNA Directed PoL V Subunit1)—DRD1 (Defect in RNA Directed DNA Metrology 1) in the RdDM pathway [27]. Although it was reported that distant hybridization between *Raphanus sativus* and *Brassica alboglabra* induced rRNA gene silencing [28], there was no direct evidence for proving transgene silencing could be happened in backcross progenies between transgenic crops and their wild relatives.

In this research, we aim to (1) reveal the transmission and expression of transgenes in backcross progeny by herbicide screening and molecular biology identification, (2) illustrate the relationship between DNA methylation and gene silencing in this event, (3) clarify the adaptability of transgenic expressing and silencing progenies to the environment and potential ecological risk. Above all, the results will not only provide a more scientific and in-depth basis for the risk assessment of transgene introgression but also ideas for molecular breeding technology to cultivate transgene *Brassica napus* with low transgene flow risk.

## 2. Results

### 2.1. Transmission and Expression of Transgenes in Backcross Progenies

Self-pollinated progenies (BC1F) of the first backcross generation (BC1) between transgenic *Brassica napus* and wild *Brassica juncea* were used as research material in this study.

BC1F3, which were obtained from the self-pollination of glufosinate-resistant BC1F2 (Resistant-gene-expressing) were used as the material firstly. Through Polymerase Chain Reation (PCR) for testing *PAT* gene, the BC1F3 were separated into two types: plants carried *PAT* gene and No-resistant-gene plants (Figure 1). Plants carried *PAT* gene were treated by 18% glufosinate SL (Basta, Bayer Crop Science, Germany) at 700 g a.i. ha^−1^. Finally, all plants (BC1F3) were separated into three types: resistant-gene-expressing (R), resistant-gene-silencing (S) and no-resistant-gene (NR) plants (Figure 2).

Three types of BC1F4 plants, which are resistant-gene-expressing (RR), resistant-gene-silencing (RS) and no-resistant-gene (NR) were obtained from the self-pollination of glufosinate-resistant BC1F3 (R). Meanwhile, three types of BC1F4 plants, which were resistant-gene-expressing (SR), resistant-gene-silencing (SS) and no-resistant-gene (NR) were obtained from the self-pollination of resistant-gene-silencing BC1F3 (S).

Three types of BC1F5 plants, namely, resistant-gene-expressing (RRR), resistant-gene-silencing (RRS) and no-resistant-gene (NR) were obtained from the self-pollination of glufosinate-resistant BC1F4 (RR). Meanwhile, three types of BC1F5 plants, which were resistant-gene-expressing (RSR), resistant-gene-silencing (RSS) and no-resistant-gene (NR) were obtained from the self-pollination of resistant-gene-silencing BC1F4 (RS).

Three types of BC1F5 plants, which were resistant-gene-expressing (SRR), resistant-gene-silencing (SRS) and no-resistant-gene (NR) were obtained from the self-pollination of glufosinate-resistant BC1F4 (SR). Meanwhile, three types of BC1F5 plants, which were resistant-gene-expressing (SSR), resistant-gene-silencing (SSS) and no-resistant-gene (NR) were obtained from the self-pollination of resistant-gene-silencing BC1F4 (SS).

It showed the trait of segregation occurred in the progenies of different phenotypes, and the resistance segregation ratio was close to 1:1 in each generation ([Fig plants-11-03187-ch001]).

From the above results, it was found that more than 80% of the plants carried thetransgene in BC1F3 and BC1F4, and more than 90% in BC1F5, showing a high level of transgene transmission, even though the X^2^ text value showed significant differences compared with the theoretical value (Table 1). However, the proportions of survival plants under glufosinate selection were nearly 50%, showing a significant difference with the percentage of plants carrying the transgene (Table 1). The result demonstrated that the glufosinate-resistant gene was silenced in approximately 50% of plants of the carried transgenes BC1F3, BC1F4 and BC1F5. Gene silencing occurred in each generation of backcross progenies, whatever the previous generation plant was resistant expressing or silencing plants. This result indicated that transgene silencing in this event was inherited to progenies in the current experimental conditions.

### 2.2. The mRNA Expression of the PAT Gene in Resistant-Gene-Silencing Plants Was Significantly Lower Than That in Resistant-Gene-Expressing Plants

The results of the expression of *PAT* gene mRNA by qPCR showed that the mRNA expression of the transgene *PAT* in resistant-gene-silencing plants was significantly lower than that in resistant-gene-expressing plants, and the expression level did not varied significantly at different growth stage (Figure 3), and each generation showed the same results. It indicated that the herbicide-resistant transgene was silencing in backcross progenies.

### 2.3. PAT Possesses the Same Insertion Site in Backcross Progenies with That in Brassica napus

We obtained about 900 bp flanking sequence (Figure 4), which verified that the insertion sites of resistant-gene-expressing and resistant-gene-silencing plants in the first generation of backcross were consistent with those of glufosinate-resistant transgenic *Brassica napus.*

### 2.4. Methylation at the Promoter Was an Important Factor Leading to Gene Silencing

The results are shown in Table 2. In the resistant-gene-expressing plants, the number of methylation sites in the promoter region of the *PAT* gene was 0, while in the resistant-gene-silencing plants, the number of methylation sites in the promoter region of the *PAT* gene was 6 to 9, which was significantly higher than that in the resistant-gene-expressing plants. Among the all types of plants selected in the experiment, there were 3 to 6 methylation sites in *PAT* gene sequence, and there was no significant difference. Based on the results of *PAT* gene expression, it is speculated that the silencing of the resistance gene *PAT* in the self-pollinated progenies of the first backcross generation between glufosinate-resistant transgenic *Brassica napus* and *Brassica juncea* may be caused by the methylation of the promoter region of *PAT* gene.

### 2.5. 5-azaC Suppresses the Occurrence of Gene Silencing

The results showed that a 250 µmol/L 5-azaC pretreatment of BC1F3 progeny seeds (BC1F4) significantly increased the proportion of resistant-gene-expressing plants (Figure 5 and Table 3), indicating that 5-azaC reduced the proportion of resistant-gene-silencing plants by reducing the frequency of methylation at the promoter of the *PAT* gene. It was confirmed from another point of view that methylation of *PAT* gene promoter sequence was an important factor leading to resistant gene silencing.

### 2.6. Backcross Progenies Showed Similar Fitness with Wild Brassica juncea

BC1F3, BC1F4 and BC1F5 showed similar fitness-associated traits and composite fitness compared with wild *Brassica juncea*. Meanwhile resistant-gene-silencing and resistant-gene-expressing plants also displayed similar fitness between the same self-pollinated generations (Table 4, Table 5 and Table 6).

## 3. Discussion

The cultivation and commercial application of herbicide-resistant crops will be an important trend in agricultural development. However, the problem of transgene flows into their relatives is an important factor that hinders the commercialization of herbicide-resistant crops. Therefore, it is of practical significance to cultivate crops with low risk of transgene introgression into weeds. Assuming that the transgenes are silenced after flowing from HR crops to their relatives via pollen, these progenies will die after applying the target herbicide in the HR crop field, which will help to reduce the risk of herbicide resistance gene introgression into weeds and the difficulty of weed control.

Our previous research demonstrated that the survival rate under herbicide selection, and fitness of the herbicide resistant progenies between wild *Brassica juncea* (AABB) and herbicide resistant transgenic *Brassica napus* (AACC) with the transgene on the C-chromosome were significantly lower than those on the A-chromosome [14]. The current studies have found that there was transgene silencing in the backcross progenies of transgenic *Brassica napus* and wild *Brassica juncea* which transgene was located on the C-chromosome, and the resistant-gene-expressing and resistant-gene-silencing progenies have been segregated from the self-pollinated progenies of the plants with resistance gene silencing. Although the progenies of these two different phenotypes showed similar fitness-associated traits and composite fitness without herbicide selection, plants that cannot normally express the transgene will be killed under the pressure of herbicide selection, while plants that can stably express the transgene will survive and reproduce stably, and have the possibility of becoming new species in nature, for the selection pressure is one of the driving forces of species evolution [29].

Chromosomal abnormalities are usually found in heterochromatin regions where the gene sequence is highly methylated. If the transgene was integrated into a highly methylated region or a heterochromatin region with low transcriptional activity, it couldn’t be stably expressed, showing a positional effect at the transcriptional level. In the heterochromatin region, the gene sequence is often highly methylated and cannot be normally expressed, and the methylation at the gene promoter is an important feature of gene silencing.

The insertion of the transgene changes the original CG content of the chromosome, and the recipient plant will recognize the invading gene by the change of CG content, and finally lead to gene silencing caused by heterochromatin [30,31,32]. This is a self-defense mechanism of organisms against foreign genes, which is summarized in a survival strategy. In our previous research, karyotyping and fluorescence in situ hybridization–bacterial artificial chromosome (BAC-FISH) analyses showed that backcross progenies from the cultivars with transgenes located on either A- or C-chromosome were mixoploids. However, we found that chromosome pairing of pollen mother cells was more irregular in progenies from cultivar whose transgene located on C- than on A-chromosome [14]. Furthermore, Simple Sequence Repeat (SSR) analysis showed that two to three markers (Specific fragments of C5, C6 and C7 chromosome) of C chromosome had been lost after four generations of self-pollination of BC1, indicated that the mixoploid resulted in the cytological instability, that is, the heterologous C chromosome which carried PAT could induced heterochromatin and finally caused transgene silencing [14]. At the same time, the C5, C6 and C7 chromosome showed low genetic stability, which provides a basis for the future transgenic target location in genome of transgenic *Brassica napus*, that is, we could design the insertion site of foreign genes on the C chromosome, especially on these three chromosomes to reduce the probability of transgene introgression from transgenic crops to their relatives, thereby reducing the potential ecological risk.

Gene silencing at the transcriptional level is mainly caused by copy number [33,34], positional effect [30], DNA methylation of transgene promoter [35,36], homologous sequence co-inhibition and heterochromatization [32], as well as transgene damage and deletion. Methyltransferases recognize structures formed by spontaneous pairing between multiple copies of repeats, thereby inhibiting its expression [33,34,37]. This kind of gene silencing is similar with the repeat induced point mutation (RIP) found in fungi [38]. Under normal conditions, it is necessary to study the copy number of foreign genes and the genetic law of the copy number in progenies in future research. But in our study, if the gene silencing was caused by the copy number of *PAT*, gene silencing would also occur in the progenies of the paternal self-pollinated progenies. However, we have never found that the self-pollinated progenies of the paternal parent (*Brassica napus*) carried resistance genes but did not express them, that is, the inheritance of the self-pollinated progenies of the paternal parent was stable. Therefore, we speculate that the occurrence of gene silencing in this event may not be caused by the copy number of *PAT*.

DNA methylation plays an important role in regulation of transgene expression in transgenic crops, usually leads to gene silencing [39]. In addition, environmental factors such as elevated temperatures can also regulate gene expression by accelerating the accumulation of DNA mutations in plants [40]. Studies have shown that methylation at transgenic promoter (such as CaMV35s) may cause silence of gene, and methylation in the coding region shows no or little effect on gene expression [41,42,43]. In conclusion, there is a positive correlation between gene promoter methylation and gene silencing [44]. In our research, through the detection of transgenic methylation sites, it was found that the degree of methylation in the promoter (CaMV35s) and *PAT* sequence was different in resistant-gene-expressing and resistant-gene-silencing plants. In resistant-gene-expressing plants, the promoter region did not undergo methylation, while in resistant-gene-silencing plants, the number of methylated cytosine sites in the promoter region was 6 to 9, and that was the reason for the glufosinate resistance gene had been silenced in the backcross progenies.

Meanwhile, DNA demethylation, stands in the opposition of DNA methylation, also plays an important role in the regulation of gene expression. 5-azaC, a kind of methyltransferase inhibitor, plays an important role in regulating the growth and development of plants through active demethylation [45,46]. In our study, we found that a 250 µmol/L 5-azaC pretreatment seeds of BC1F3 (BC1F4) significantly increased the proportion of resistant-gene-expressing plants, indicating that 5-azaC reduced the probability of resistance gene silencing by reducing the frequency of methylation at the promoter of the *PAT* gene. It was confirmed from the anther angle that methylation of *PAT* gene promoter sequence was an important factor leading to resistance gene silencing.

The flanking sequence of PAT gene promoter CaMV35s had not changed in the progenies compared with transgenic Brassica napus. However, we designed probes with homologous fragments of *Brassica napus* as tags to capture signals and perform qualitative analysis on 942 bp fragments but not the whole genome. This did not rule out the possibility that these fragments (promoter flanking sequences) were homologously paired with A chromosome during meiotic chromosome pairing and thus become a part of A chromosome. Therefore, the existing experimental results couldn’t verify whether the flanking sequence was accurately located on the C-chromosome. Precise gene localization, for example, in situ hybridization experiments, will become the key to the experimental process.

At the same time, we should not ignore the possible regulatory role of methyltransferase and demethylase in gene silencing. In plants, METHYLTRANSFERASE 1 *(MET1*) is mainly responsible for CG methylation, CHROMOMETHYLASE 3 (*CMT3*) is mainly responsible for CHG methylation, and DOMAINS REARRANGED METHYLASE 1 (*DRM1*) and DOMAINS REARRANGED METHYLASE 2 (*DRM2*) are mainly responsible for CHH methylation [47,48,49,50,51]. These methylation modifications are regulated by the RdDM pathway. In addition, DNA demethylase regulates the occurrence of active demethylation and passive demethylation. Passive demethylation refers to that after DNA replication, the newly synthesized DNA strand cannot be methylated due to the inactivation of the maintenance DNA methyltransferase. Active demethylation does not depend on DNA replication and occurs under the catalytic conditions of a series of enzymes, such as REPRESSOR OF SILENCING 1 (*ROS1*), TRANSCRIPTIONAL ACTIVATOR DEMETER (*DME*) and *DME* LIKE PROTEIN (*DML*) [52], in which *ROS1* mainly mediates the demethylation of gene promoter regions. In the future, the cloning and detection of methylation- and demethylation-related enzymes in backcross progenies may provide us with more molecular biological evidence of the mechanism of gene silencing.

In summary, we revealed the occurrence of gene silencing in this event, and clarified the possible causes and mechanisms of its occurrence to a certain extent through molecular biological means. However, further in-depth study is still required. The gene silencing frequency in this event was nearly 50%, which seemed to be in a ‘dynamic equilibrium’ state, and it could stably emerge in progenies, which was extremely rare. Therefore, the author believed that the progenies in this research can also be used as a special model for the study of gene silencing, providing a new thinking and material basis for the study of molecular biological regulation or genetic regulation related to gene silencing. With the development of gene editing technology in recent years, gene insertion and location are becoming more and more accurate. Inserting foreign genes into the designated region of the C-chromosome with low risk has considerable guiding significance for the selection of targeted insertion sites in transgenic crops. Our research provided a more scientific and in-depth basis for the risk assessment of the flow and infiltration of the resistance gene of herbicide-resistant transgenic *Brassica napus* to related weeds, and it also supplied ideas for molecular breeding technology to cultivate herbicide-resistant *Brassica napus* with low resistance gene flow risk to weeds.

## 4. Materials and Methods

### 4.1. Plant Materials

Glufosinate-resistant transgenic *Brassica napus* (Liberty link, event HCN28) came from Canada, contains a homozygous *PAT* gene. Wild *Brassica juncea* was collected in Jiangpu, Nanjing. Song Xiaoling developed BC1 of *Brassica napus* and wild *Brassica juncea*, the detailed process was described by Song et al. [14]. Crossing scheme of hybridization and backcrossing between wild *Brassica juncea* and transgenic glufosinate-resistant *Brassica napus* was showed in Appendix A based on the research of Song Xiaoling et al. [14].

### 4.2. Biochemical Reagents and Instruments

18% glufosinate soluble agent (Baosteel, Bayer Crop Sciences, Germany), BU Taq 2× Master PCR mix (purchased from Baosheng biological company), PMD ™ 19-T ligase and methylation Kit, Master PCR mix, X-gal, IPTG, ampicillin and other reagents are all purchased from Nanjing Tianwei Co., Ltd., Nanjing, China.

### 4.3. Plasmid and Competent State

The cloning vector was PMD ^TM^ 19-T and *E. coli* DH5α competent (purchased from Takara Baosheng Biological Company, Dalian, China).

### 4.4. Analysis Software

Bioedit, BioXM 2.6 and NCBI primer design software and sequence alignment software.

### 4.5. Selection for Tolerance to Herbicide and Fitness-Associated Traits in the Backcross Progeny

Described of plants’ grown and placed was shown in Supplementary Methods 1.1. Method of DNA template extraction was shown in Supplementary Methods 1.2. Design of *PAT* primers (Appendix A), system of PCR experiment (Appendix A) were as described by Song et al. [14].

Through PCR for testing PAT gene (Primers for PAT were showed in Appendix A), the progenies were separated into two types: plants carried PAT gene and No-resistant-gene plants. Leaf of plant carried PAT gene were treated by 18% glufosinate SL (Basta, Bayer Crop Science, Germany) at 700 g a.i. ha^−1^. If the leaf withered, these plants were considered as resistant-gene-silencing (S), If the leaf was normal, the plant was considered resistant-gene-expressing (R). Then these two types of plants were sprayed by glufosinate as the methods described by Song et al. [14] to verify the accuracy of the screening results. Thus all plants were separated into three types: resistant-gene-expressing (R), resistant-gene-silencing (S) and no-resistant-gene (NR) plants, were obtained from the self-pollination of glufosinate-resistant BC1F2 (Re-resistant-gene-expressing) (Figure 1).

### 4.6. Amplification of DNA Flanking Sequences of Glufosinate-Resistant Transgenic Brassica juncea Transformants (Tail PCR)

According to the promoter (CaMV35s) of the glufosinate-resistant gene (*PAT*) and the *Brassica napus* gene sequence on the left side of the transgene, primers were designed with the help of Bioxm 2.6 software in Appendix A [14]. See Appendix A for the PCR reaction system. The reaction conditions of PCR system was shown in Supplementary Methods 2.1.

### 4.7. 5-azaC Treatment Testing

100 seeds were placed in Petri dishes containing filter paper which was saturated with different concentration 5-azaC solution or double distilled water (ddH_2_O), respectively. Then these Petri dishes were placed randomly in a 4 °C refrigerator. The filter paper saturated with treatment solution was replaced every 24 h for 5 consecutive days. The 5-azaC treated seeds and the ddH_2_O treated seeds were sown at the same time and cultured in the same condition as the method described in supplementary methods 1.1 and 5.1. Finally, the concentration of 250 µmol/L of 5-azaC was used for experiment due to higher emergency percentage and survival rate after glufosinate application as the method described in Materials and Methods Section 4.5.

### 4.8. RNA Extraction

When the tested plants grew to the 4–6 leaf stage, 10 resistant expressing and resistant non-expressing plants of different backcross progenies were sprayed with glyphosate. Leaves were taken and then frozen in liquid nitrogen and stored in a −80 °C refrigerator.RNA extraction was carried out according to the instructions of the RNA Extraction Kit Biospin Plant Total RNA Extraction Kit of Bioflux. The specific operating procedure was shown in Supplementary Methods 3.1.

### 4.9. cDNA Synthesis

According to the Reverse Transcriptase Primescript ™ RT reagent kit with gDNA eraser (perfect real time) (Takara company) instructions to prepare reaction solution. Preparation of the reverse transcription master mix of *PAT* genes were shown in Appendix A. Reaction system of the reverse transcription master mix of PAT genes were shown in Appendix A. Reaction solution were shown in Supplementary Methods 3.2.

### 4.10. qPCR Testing

The primer design was shown in Appendix A. PCR programs were according to the Reverse Transcriptase Primescript™ RT Reagent Kit with gDNA Eraser (perfect real-time) (Takara company) instructions in Appendix A. qPCR amplification reaction system of *PAT* genes was shown in Appendix A.

### 4.11. Methylation Site Detection

DNA methylation was performed by bisulfite treatment according to the instructions of Methyldetectortm Bisulfite Modification kit. Experimental process of DNA bisulfite treatment was shown in Supplementary Methods 4.1.

According to the promoter (CaMV35s) and *PAT* gene sequence, the designed primers are shown in Appendix A. See Appendix A for PCR reaction system.

Detailed process of PCR amplification of promoter (CaMV35s) and PAT gene was shown in Supplementary Methods 4.2.

In the DNA sequence, if cytosine (C) is methylated, the methylated cytosine will not be converted to Uracil (U) under sulfite treatment but still exists in the gene sequence as cytosine (C). The transformants were sent to a Shanghai biotechnology company for sequencing. With the aid of the analysis software Bioedit, the sequence was compared with the original sequence of the transgene (CaMV35s and *PAT* gene) to find the location and number of cytosine (C) in the measured sequence. Because there are certain errors in the sequencing results, it is considered that the same methylation site appears twice or more in the three replicates of each plant.

### 4.12. Fitness Analysis

Planting methods were shown in Supplementary Methods 5.1.

The fitness associated traits measured in vegetative growth stage were dry above-biomass, plant height, stem diameter, number of effective branches. The fitness-associated traits measured in reproductive growth stage are Number of silique/plant, seed weight, silique length, and seed number/silique. See Appendix A for the measurement methods.

Calculation method of composite fitness was described in Supplementary Methods 5.2 by Song et al. [14].

### 4.13. Data Statistics

The Tukey test in SPSS 20.0 statistical software was used for data statistical analysis. The variance significance analysis was carried out for the backcross progeny of glufosinate-resistant transgenic *Brassica napus* and wild *Brassica juncea*. The traits and composite fitness between wild *Brassica juncea* and each backcross progeny were compared by one-way ANOVA/A.

## 5. Conclusions

Transgene silencing occurred in the backcross progenies between glufosinate-resistant *Brassica napus*, carrying *PAT* gene which located on C-chromosome, and wild *Brassica juncea* in a near 50% frequency. Six to nine DNA methylation sites were found at the promoter of these transgene silencing plants, and 5-azaC treatment on seeds effectively inhibited the occurrence of transgene silencing, which clarified that DAN methylation at the promoter of the *PAT* gene was an important factor leading to transgene silencing in this study.

## Data Availability

The data are available upon request from the corresponding author.

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
