# Peer review of "Transgene Was Silenced in Hybrids between Transgenic Herbicide-Resistant Crops and Their Wild Relatives Utilizing Alien Chromosomes"

_plants, 2022, doi:10.3390/plants11233187_

Round 1
Reviewer 1 Report
This work by Shao et al.reports on the occurrence of transgene silencing events in backcross progenies between transgenic crop utilizing alien chromosomes and its wild relatives, and the methylation at the gene promoter as an important factor leading to gene silencing. The work is generally meaningful, and based on my understanding, in addtion to English editing throught the article, the specific comments are given as follows:
Q1: in Abstract, …“gene silencing was happened…” to “gene silencing occurred”
Q2: in Keywords, please put keywords in an alphabetical order.
in Introduction part,
Q3: In the third paragraph, the author mentioned that stable inheritance and expression of HT transgenes in hybrids is considered as a necessary condition for transgenic introgression. Is there literature support?
Q4: Fewer references, such as the relationship between methylation and gene silencing.
Results,
Q5: First, the author's description of the material is less. Figure 1, this layout is not appropriate, not easy to distinguish between plant materials, and it is difficult to see plant details. Similarly, authors should add plant photos to the supporting material.
Q6: As far as I know, transgenic material need at least three replicates.
Q7: Should PCR results be used as supporting files?
Q8: Could the authors clarify this PAT gene better?
Q9: What does BC1 mean in the article? Authors should express clearly.
Q10: The authors believe that promoter methylation is an important factor leading to gene silencing. Have other experiments to prove this result? That way, your results would be more accurate.
Q11: Authors should explain why to use 5-azac, is there any other way? And authors should add plant photos after using 5-azac.
Discussion,
Q12: The discussion section resembles the background. And there is not much discussion about the results of this article. Authors should compare and discuss this article with related research on other species.
Q13: Authors should discuss the focus of this article in depth, not all views should be discussed.
Q14: Conclusion should not be included in the discussion, authors need to draw a conclusion.
Author Response
Thank you for reviewing and commenting on the article ‘Transgene was Silenced in hybrids between Transgenic Herbicide-Tolerant Crops and Their Wild Relatives Utilizing Alien Chromosomes’ in your busy schedule. I will reply to your questions and suggestions as follows:
Q1&Q2:
in Abstract, …’gene silencing was happened…’ to “gene silencing occurred”
in Keywords, please put keywords in an alphabetical order.
Thanks for your modification suggestions. The abstract and keywords has been modified according to your comments.
in Introduction part
Q3: In the third paragraph, the author mentioned that stable inheritance and expression of HT transgenes in hybrids is considered as a necessary condition for transgenic introgression. Is there literature support?
Thanks for your questions. We apologize for the careless omission of the manuscript. The paragraph of ‘Stable inheritance and expression of HT transgenes in hybrids is considered as a necessary condition for transgenic introgression…’ is supported by 5 references but not marked in the original manuscript, and the references have been supplemented at present.
Q4: Fewer references, such as the relationship between methylation and gene silencing.
Thanks for your suggestions. Not only a number of references have been supplemented and added as required, but we also supplied the discussion on the mediating mechanism of different gene silencing ways. At the same time, we also had a more in-depth discussion on methylation and gene silencing in ‘Discussions’ section.
Q5: First, the author's description of the material is less. Figure 1, this layout is not appropriate, not easy to distinguish between plant materials, and it is difficult to see plant details. Similarly, authors should add plant photos to the supporting material.
Thanks for your modification suggestions. We have re-uploaded clearer pictures, and added photos of experimental plants in each growth stages in Supplementary Graphs (Graph 2) based on your suggestions. Meanwhile, We have added the information of materials in the first part of Supplementary Meterials.
Q6: As far as I know, transgenic material need at least three replicates.
Thanks for your questions. Sorry for the omission of the plants information again. For the backcross progenies in this study, 10 mother plants in each generation that displayed the highest seed fertility for each lineage were chosen, and at least 50 filled seeds from each of the ten mother plants were selected randomly. That means we do a replication of 10 in our study. We have added the information of materials in the first part of Supplementary Materials.
Q7: Should PCR results be used as supporting files?
Thanks for your questions. The PCR result of pat gene is an important basis for gene silencing, that is, it carries resistance gene but didn’t show herbicide resistance. The PCR results of insertion site proved that the insertion site of pat gene did not change in progenies, so the influence of insertion site on gene silencing doesn’t need to be discussed. Both results are very important to the integrity of the logic of the article, so we suggested that the PCR results are more appropriate in the ‘Result’ section.
Q8: Could the authors clarify this PAT gene better?
Thanks for your questions. Yes, we have added the introduction of pat gene ‘PAT is the type of N-acetyltransferase gene which has the detoxification effect of glyphosate. Bayer Crop Science cloned the genes bar and PAT that can encode N-acetyltransferase from Streptomyces hygroscopicus and Streptomyces viridochromo respectively’ to the ‘Introduction’ section.
Q9: What does BC1 mean in the article? Authors should express clearly.
Thanks for your questions. Sorry for our negligence, BC1 means Self-pollinated progenies of first backcross generation between transgenic Brassica napus and wild Brassica juncea (BC1F), we have added it to the top of ‘Results’.
Q10: The authors believe that promoter methylation is an important factor leading to gene silencing. Have other experiments to prove this result? That way, your results would be more accurate.
Thanks for your questions and suggestions. In order to prove the reliability of the theory, we designed an experiment to treat seeds with methyltransferase inhibitor 5-azac. It was found that the frequency of gene silencing decreased significantly when methylation was inhibited by 5-azac, so on the other side it was proved that methylation was an important factor leading to gene silencing of experimental materials.
Q11: Authors should explain why to use 5-azac, is there any other way? And authors should add plant photos after using 5-azac.
Thanks for your questions and suggestions. Sorry for our negligence, we used to put the photos on but somehow they didn't appear in the manuscript when we submitted. Now we have re-added 5-azac photos to ‘Results’ section. As stated in the above question, 5-azac, as a methyltransferase inhibitor which inhibited methylation, significantly reduced the frequency of gene silencing, showed that methylation is an important factor in the occurrence of gene silencing in this experiment.
Q12&13:
The discussion section resembles the background. And there is not much discussion about the results of this article. Authors should compare and discuss this article with related research on other species.
Authors should discuss the focus of this article in depth, not all views should be discussed.
Thanks for your modification suggestions. We considered that your opinion is correct. We have modified the ‘Discussion’ section and showed an in-depth discussion in our research and added the ‘Conclusion’ section as required. In the second and the third paragraph of the Discussion, we discussed the effect of chromosomes on gene silencing, as we speculate that promoter methylation and gene silencing are the result of events, and the basic mechanism of promoter methylation might be the chromosome abnormalities caused by the recognition of heterologous chromosomes. Although not introduced in the results, we think it is necessary to discuss them. In addition, due to less genetic research on the foreign genes located in the C chromosome of transgenic oilseed rape and its relatives, we prefer to compare and discuss with this experiment with relevant experiments on oilseed rape and its relatives, such as Brassica juncea or Brassica rapa. Of course, Arabidopsis thaliana, a model plant with high reference, is also in our scope of discussion, such as methylation related research. We speculated on the possible causes of gene silencing, such as the possibility of DNA methylation induced by heterochromatin caused by heterochromosomes, or gene silencing caused by gene copy number, led to the accuracy and rationality analysis of the results in this paper, and expounded the research route and research value of the results in the future research on transgenic oilseed rape.
Q14: Conclusion should not be included in the discussion, authors need to draw a conclusion.
Thanks for your suggestions. We have rewroted the section of Conclusions and modified section of the Discussion as requested.

Reviewer 2 Report
The manuscript Transgene was Silenced in hybrids between Transgenic Herbicide-Tolerant Crops and Their Wild Relatives Utilizing Alien Chromosomes by Zicheng Shao , Lei Huang , Yuchi Zhang , Sheng Qiang , Song Xiaoling describes a study on herbicide resistance inheritance and its association with methylation. The work contains all the necessary parts and reflects the issue, however, some negligence in the design needs to be corrected.
So in materials and methods, the phrase is surprising: The results demonstrated that the glufosinate-resistant gene was silenced in approximately 50% of plants of carried transgenes BC1F3 and BC1F4, BC1F5. What is this data doing in this section? If this is the result, then its place in the results, if the old data - then where is the link?
It is not at all clear how many plants were taken per sample.
How long have they been grown?
The presence of two figures 1 and figures 2 is surprising. It is obvious that the authors did not carefully read the manuscript before sending ...
Figure 2 first contains *** - but what they refer to is not clear.
Also, for example, figure two two is clearly embarrassing. The arrows do not clearly correspond to what, the signature does not contain complete information about the method by which the image was obtained, while this is mandatory information.
The conclusion of the manuscript should be singled out in a separate paragraph.
Authors are advised to read the manuscript carefully before resubmitting.
Author Response
Thank you for reviewing and commenting on the article ‘Transgene was Silenced in hybrids between Transgenic Herbicide-Tolerant Crops and Their Wild Relatives Utilizing Alien Chromosomes’ in your busy schedule. I will reply to your questions and suggestions as follows:
Q1: So in materials and methods, the phrase is surprising: The results demonstrated that the glufosinate-resistant gene was silenced in approximately 50% of plants of carried transgenes BC1F3 and BC1F4, BC1F5. What is this data doing in this section? If this is the result, then its place in the results, if the old data - then where is the link?
Thanks for your questions. We have checked the manuscript and were sure that the paragraph ‘The results demonstrated that the glufosinate-resistant gene was silenced in approximately 50% of plants of carried transgenes BC1F3 and BC1F4, BC1F5’ is in the Results section rather than the Materials and Methods section.
Q2&3:
It is not at all clear how many plants were taken per sample.
How long have they been grown?
Thanks for your questions. We used least 50 filled seeds from each of the ten mother plants were selected randomly and each seed sown directly into individual plastic pots. The plant growth cycle is 6-7 months. We have added specific planting methods and details to the first paragraph of Supplementary Materials document. Since our experimental materials have been introduced in detail in the papers previously published by our laboratory [14], they are not repeated in this paper for brevity.
Q4&5:
The presence of two figures 1 and figures 2 is surprising. It is obvious that the authors did not carefully read the manuscript before sending ...
Figure 2 first contains *** - but what they refer to is not clear.
Thanks for your modification suggestions. Sorry for our negligence, and we have renumbered all figures and tables as required and made sure they are correct as required. We apologize for the inconvenience caused to your review。
Q6: Also, for example, figure two two is clearly embarrassing. The arrows do not clearly correspond to what, the signature does not contain complete information about the method by which the image was obtained, while this is mandatory information.
Thanks for your modification suggestions. Sorry for our negligence again and we have remade these figures for better display.
Q7: The conclusion of the manuscript should be singled out in a separate paragraph.
Thanks for your modification suggestions. We have rewrote the section of Conclusions and modified section of the Discussion as requested.
Q8: Authors are advised to read the manuscript carefully before resubmitting.
Thanks for your suggestions. We sincerely apologize again for troubles caused by the urgent submission of manuscript, we are responsibility for these negligence. We have seriously re-proofread and modified the manuscript.

Reviewer 3 Report
It is a good study that contributes to an important question--if there is safe site for editing? With this information the precise gene editing tech could help to make safe manipulation. This provides an optimistic opportunity for the future of genetic engineering.
Then I want to talk some editorial aspects. Perhaps the editor may also note these issues. It seems that this ms has some supplementary materials, which has to be mentioned in the end of the paper. Where these supplementary materials will be located.
In the Data Availability Statement part, I guess that the authors might make wrong statement, where they indicated the sources of the sequences they used other than the place they store their study data.
A small thing is that the scientific names of the species Brassica napus and B. juncea shall be in italics. Some of them are, but please be consistent through the whole ms.
Now come to the academic issues:
Could the authors clarify in the text (discussion) if the methylation could be a general phenomenon during gene introgression between B. napus and B. juncea or it is only applicable to their case? Same question on the inserted foreign transgenes, if it is specific for the pat gene (their transgenic event only) or any other transgenes that inserted into C-genome. If there is a specific literature for the glufosinate resistant transgenic B. napus would helpful.
Regarding fitness, in the agricultural field, when additional commercialization of herbicides resistant crops released, more herbicides application could be anticipated. If there is herbicide application in the environment, we cannot say “there was no significant difference in fitness-associated traits and fitness component between resistant-gene-expressing plants and resistant-gene-silencing plants”.
The authors stated in their ms that “Precise gene localization, including genome related SSR experiments and in situ hybridization experiments will become the key to the experimental process.” I believed that these are good points. I remember there are published papers on these issues using SSR markers and/or in situ hybridization in study the safe site of insertion of transgenes in Brassica napus. Could the authors refer to these results and discuss together with the insight drawn from their own finding. That could be interesting!
the table labeling are confusing, please ask the authors check carefully, table 3-1, 3-2 (where is 3-2?) and table 4-1, 4-2.
Author Response
Thank you for reviewing and commenting on the article ‘Transgene was Silenced in hybrids between Transgenic Herbicide-Tolerant Crops and Their Wild Relatives Utilizing Alien Chromosomes’ in your busy schedule. I will reply to your questions and suggestions as follows:
Q1: Then I want to talk some editorial aspects. Perhaps the editor may also note these issues. It seems that this ms has some supplementary materials, which has to be mentioned in the end of the paper. Where these supplementary materials will be located.
Thanks for your modification suggestions. Sorry for our negligence, and we have added Supplementary Materials as an extra Word document to our submission.
Q2: In the Data Availability Statement part, I guess that the authors might make wrong statement, where they indicated the sources of the sequences they used other than the place they store their study data.
Thanks for your modification suggestions. We considered that your opinion is correct, and we have modified the Data Availability Statement already.
Q3: A small thing is that the scientific names of the species Brassica napus and B. juncea shall be in italics. Some of them are, but please be consistent through the whole ms.
Thanks for your modification suggestions. Sorry for our negligence. We have modified and unified the species names in manuscript as required.
Q4: Could the authors clarify in the text (discussion) if the methylation could be a general phenomenon during gene introgression between B. napus and B. juncea or it is only applicable to their case? Same question on the inserted foreign transgenes, if it is specific for the pat gene (their transgenic event only) or any other transgenes that inserted into C-genome. If there is a specific literature for the glufosinate resistant transgenic B. napus would helpful.
Thanks for your suggestions. Very nice points, for pat gene silencing is really only applicable for the backcross progenies between B. napus and B. juncea. In the INTRODUCTION section, we mentioned that transgenic B.napus and its relatives such as B.rapa, the foreign genes located on the C chromosome will be more difficult to inherit stably than those located on the A chromosome [21-23]. However, there is no same type report that foreign genes are confirmed to have gene silencing in our experimental materials. We believed that this phenomenon is specific for our study. Of course, we have elaborated on this in the DISCUSSION section according to the suggestions as required.
Q5: Regarding fitness, in the agricultural field, when additional commercialization of herbicides resistant crops released, more herbicides application could be anticipated. If there is herbicide application in the environment, we cannot say “there was no significant difference in fitness-associated traits and fitness component between resistant-gene-expressing plants and resistant-gene-silencing plants”.
Thanks for your suggestions. We considered that your opinion is correct. The sentence had been modified to ‘However, the progenies of these two different phenotypes showed similar fitness-associated traits and fitness component.’ In the DISCUSSION section.
Q6: The authors stated in their ms that “Precise gene localization, including genome related SSR experiments and in situ hybridization experiments will become the key to the experimental process.” I believed that these are good points. I remember there are published papers on these issues using SSR markers and/or in situ hybridization in study the safe site of insertion of transgenes in Brassica napus. Could the authors refer to these results and discuss together with the insight drawn from their own finding. That could be interesting!
Thanks for your suggestions. Very nice points too! The article published by our laboratory last year has studied the genetic stability of the gene in the C chromosome of the same experimental material as this experiment through SSR experiment, and found that the specific segment of the C chromosome was significantly lost faster than that of the A chromosome in progenies, which indicated that the C chromosome of backcross progenies was not stable after self-pollinated [14]. Furthermore, we elaborated that two to three markers (Specific fragments of C5, C6 and C7 chromosome) for fourth generation of BC1 had been lost, indicated that the C5, C6 and C7 chromosome showed low genetic stability, which provided a basis for the future transgenic target location in genome of transgenic oilseed rape[14]. We have discussed this in the DISCUSSION section of ‘Chromosomal abnormalities are usually found…’.
Q7: the table labeling are confusing, please ask the authors check carefully, table 3-1, 3-2 (where is 3-2?) and table 4-1, 4-2.
Thanks for your modification suggestions. Sorry for our negligence. We have renumbered all figures and tables as required and made sure they are correct. We apologize for the inconvenience caused to your review.

Author Response
Thank you for reviewing and commenting on the article ‘Transgene was Silenced in hybrids between Transgenic Herbicide-Tolerant Crops and Their Wild Relatives Utilizing Alien Chromosomes’ in your busy schedule. I will reply to your questions and suggestions as follows:
Q1. 2.1 I suggested that the first three paragraphs concerning the screening experiment should be summarized in a schematic diagram, which make the readers understand clearly. This diagram can be merged into Figure 1.
Thanks for your modification suggestions. The diagram has been made and merged into Figure 1.
Q 2. Figure 1, what do these plants in the right panel represent? And the Figure 1 is not clear enough, it should be replaced by a high-resolution image.
Thanks for your questions and modification suggestions. We have already replaced the a high-resolution picture and remade the Figure 1.
Q3. The title “Table 3” under Result 2.5 section is incorrect, please renamed the “Table 3” by “Table 4”, then it should be cited in Result 2.5. In addition, all figures and tables cited throughout the manuscript should be checked carefully. For example, Table 3 instead of Table 3-1, Fig.2 instead of Fig.2-1. Table 4 should be renamed to Table 5.
Thanks for your modification suggestions. Sorry for our negligence. We have renumbered all figures and tables as required and made sure they are correct. We apologize for the inconvenience caused to your review。
Q4. 2.6. Please notice the Latin name of species and abbreviation, such as B.juncea,
Brassica juncea. Please check them throughout the manuscript.
Thanks for your modification suggestions. Sorry for our negligence. We have modified and unified the species names in manuscript as required.
Q5. 5. The last sentences in the Result section are the note of Table rather than the results. The sentence “Different lowercase letters indicate the significant difference in relative fitness levels (BC1 fitness index /Brassica juncea fitness index), p<0.05,
n=15” should be deleted because no lowercase used in Table.
Thanks for your modification suggestions. Sorry for our negligence , the last sentences in the Result section has been reset in the note.
The sentence has been deleted as required.
Q6. Suggest the “Graph 1” move to the result section 2.1 or method section.
Thanks for your modification suggestions. Graph 1 has been moved to Result section.
Q7. For qPCR testing, the description of method is not adequate, more description is needed.
Thanks for your modification suggestions. We have added the detailed qPCR experiment scheme to the 3rd part ‘qPCR analysis’ in Supplementary Materials.
Q8&9. Discussion, change “cmt3” to “CMT3”.
Discussion, in addition to DNA methylation affecting PAT gene expression, some other factors such as the occurrence of DNA mutations (spontaneous or environmental induced) during hybridization and backcrossing also may influence the expression of herbicide resistant-related genes. Thus, I suggest that the potential role of environment-induced DNA mutations (e.g., temperatures) in affecting gene expression should be discussed in the Discussion.
Thanks for your modification suggestions. Sorry for our negligence, the wrote of cmt3 has been changed to CMT3.
A wonderful point of view, we have added it in ‘Discussion’ section and the article ‘Genome-wide dna mutations in arabidopsis plants after multigenerational exposure to high temperatures’ has been referenced.

Round 2
Reviewer 2 Report
The manuscript "Transgene was Silenced in hybrids between Transgenic Herbicide-Tolerant Crops and Their Wild Relatives Utilizing Alien Chromosomes" by Zicheng Shao, Lei Huang , Yuchi Zhang, Sheng Qiang , Song Xiaoling has been substantially revised and formatted according to the rules. Technical comments have been removed. The materials are supplemented with the required data. After verification, the manuscript may be published in its present form.